# Effects of Oregano Essential Oil and/or Yeast Cultures on the Rumen Microbiota of Crossbred Simmental Calves

**DOI:** 10.3390/ani14243710

**Published:** 2024-12-23

**Authors:** Ting Liu, Zhihao Luo, Tao Zhang, Huan Chen, Xuejiao Yi, Jiang Hu, Bingang Shi, Yuxi An, Changze Cui, Xiangyan Wang

**Affiliations:** 1College of Animal Science and Technology, Gansu Agricultural University, Lanzhou 730070, China; lzhxxx0104@163.com (Z.L.); ltsecjpsktgu7@163.com (T.Z.); 17723395157@139.com (H.C.); yxj18794832346@163.com (X.Y.); shibg@gsau.edu.cn (B.S.); 15379761918@163.com (Y.A.); cuichangze0120@163.com (C.C.); wxy9242022@163.com (X.W.); 2Linxia Beef Industry Development Research Institute, Linxia 731100, China; 3Gansu Key Laboratory of Herbivorous Animal Biotechnology, Lanzhou 730070, China

**Keywords:** additive package, calves, oregano essential oil, rumen microorganisms, yeast culture

## Abstract

Feeding calves a mixture of oregano essential oil and yeast cultures led to increased rumen microbial richness and diversity, along with regulated relative abundances of particular species. Moreover, pathways associated with metabolism and antimicrobials were enriched. The research indicates that this mixed additive outperforms separate feeding of oregano essential oil and yeast culture in modulating the rumen microbial community of calves.

## 1. Introduction

Newborn calves play a crucial role in livestock farming due to their early contribution to rumen microbiota establishment, which affects feed digestion, energy conversion, growth, and later production performance and health. In light of the global antibiotic ban [1], the development of safe feed additives to enhance rumen microbiota composition has become critical. Studies show that plant essential oils, yeast cultures (YCs), antimicrobial peptides, Lactobacillus, and organic acids can serve as alternatives to antibiotics, directly benefiting calf health [2,3,4,5,6]. Oregano essential oil (OEO), a natural plant extract, is widely used as a feed additive for its antibacterial, antiviral, antifungal, and antioxidant properties [7,8,9]. Previous studies by our team demonstrated that OEO enhances calf growth and immune function [10,11,12] and modulates rumen microbiota to strengthen immunity [9]. Research also indicates that OEO increases Lactobacillus abundance in piglets [13], reduces Enterobacteriaceae, and enriches rumen cocci, bifidobacteria, and enterococci in sheep. It also boosts metabolites such as indole-3-acetic acid and indole aldehyde, improving growth and intestinal barrier function [14]. Although OEO shows promise in promoting livestock growth and refining rumen microbiota composition, most studies focus on its standalone use. Limited research addresses potential synergistic or antagonistic effects when OEO is combined with other feed additives. Thus, investigating combinations that synergistically enhance rumen microbiota in calves together with OEO remains essential.

YCs are produced by anaerobic fermentation and are subsequently dried on specific carriers under tightly controlled production conditions [15]. These include yeast cells, yeast metabolites, and components of the culture medium. YCs are widely used in animal husbandry [16]. Studies indicate that supplementing YCs can improve cattle growth performance, enhance rumen development [17,18], and increase the abundance of fiber-degrading bacteria, lactic acid-utilizing bacteria, and carbohydrate-degrading bacteria. This leads to improved rumen function and a better feed conversion rate [19,20]. Adding active yeast can accelerate the maturation of the rumen microbiota in lambs and stabilize the rumen environment [21]. In a recent study, supplementation with YCs increased the abundance of non-fiber-degrading bacteria in Tibetan sheep while reducing pathogenic bacteria in the rumen. This suggests that yeast not only supports rumen microbiota stability but also strengthens immunity [22]. In a previous study, we tested a combination of OEO and sodium butyrate but found no significant effects [9]. Given that YCs can influence the rumen microbiota of ruminants, we chose to combine yeast with OEO. Currently, the literature on the combined use of natural feed additives in ruminants is scarce, but limited studies suggest that specific combinations can improve ruminant performance. This study aims to evaluate the effects of OEO and YCs, used either alone or in combination, on the development of the rumen microbiota. The hypothesis is that combining OEO and YCs will regulate the rumen microbiota and promote gastrointestinal balance and function.

## 2. Materials and Methods

### 2.1. Test Animals

The trial was conducted at Shengze Breeding Base, Hezheng County, Linxia Prefecture, Gansu Province, China from December 2023 to May 2024. Twenty-four newborn Simmental crossbred male calves (Sire was a purebred Simmental crossed with a Simmental crossbred dam) were selected for the trial. Six disease-free calves with a birth weight of at least 35 kg were selected for each treatment. Calves were blocked by birth data and assigned to 1 of 4 treatments using a randomized complete block design (RCBD) for the 70-day experimental period. Treatments were as follows: 1. Control, (**CON**): calves fed a calf starter without additives; 2. **OEO**: calves fed a calf starter containing 60 mg OEO per kilogram body weight (BW) per day; 3. **YC**s: calves fed a calf starter containing 45 mg YC per kilogram body weight (BW) per day; 4. **MIX**: calves fed a calf starter containing OEO (60 mg/kg, BW) and YC (45 mg/kg, BW) in combination per day. The OEO was purchased as a dry granular product (Rum-A-Fresh, Ralco Nutrition, Inc., Marshall, MN, USA) containing approximately 1.3% OEO, lactic acid, cobalt carbonate, and a zeolite carrier. The YC product, containing mannan (≥20%), β-glucan (≥20%), water (≤6%), and crude protein (≤25%), was purchased from Phileo manufactured by Lesaffre (Shanghai, China).

### 2.2. Feeding Procedure

Calves were fed 3.5–4 L of colostrum within 1 h of birth, followed by 2 L 6 h later. From days 1–14, calves were allowed to suckle their dam ad libitum. On day 15, calves were placed in individual pens equipped with food bowls. Starting on day 15, calves were fed starter feed at regular intervals throughout the day. The release schedule was as follows: 2 times per day for 2 h on days 15–28, 3 times per day for 2 h on days 29–36, 3 times per day for 1 h on days 37–43, 2 times per day for 1 h on days 44–50, 1 time per day for 0.5 h on days 51–69, and 1 final release for 0.25 h on day 70, followed by weaning. Calves were provided free-choice access to starter feed and water in individual stalls. OEO and YCs were evenly mixed into the starter feed. The ingredient and nutrient composition of the calf starter is presented in Table 1.

### 2.3. Sample Collection and Measurement

On day 70 of the experimental period, 10 mL of rumen fluid was collected from each calf via a rumen suction strainer sampling method. The fluid was filtered through four layers of cheesecloth and stored in liquid nitrogen for DNA extraction and sequencing. DNA was extracted following the manufacturer’s procedures for the E.Z.N.A.^®^ Soil DNA Kit (Omega Bio-Tek, Norcross, GA, USA). The quality of extracted genomic DNA was assessed using 1% agarose gel electrophoresis, and DNA concentration and purity were determined with a NanoDrop2000 (Thermo Fisher Scientific, Waltham, MA, USA). PCR amplification of the V3–V4 variable region of the 16S rRNA gene was conducted using the upstream primer 338F (5′-ACTCCTACGGGGAGGCAGCAG-3′) and the downstream primer 806R (5′-GGACTACHVGGGTWTCTAAT -3′). Amplification was carried out on an ABI GeneAmp^®^ 9700 thermocycler (Thermo Fisher Scientific, Waltham, MA, USA). The reaction system included 4 μL of 5× TransStart FastPfu buffer, 2 μL of 2.5 mM dNTPs, 0.8 μL of 5 μM upstream primer, 0.8 μL of 5 μM downstream primer, 0.4 μL of TransStart FastPfu DNA polymerase, and 0.2 μL of 10 ng template DNA. All samples were amplified in triplicate. PCR products were extracted from a 2% agarose gel and purified with the AxyPrep DNA Gel Extraction Kit (Axygen Biosciences, Union City, CA, USA) following the manufacturer’s instructions. The purified products were quantified using a Quantus™ Fluorometer (Promega, Madison, WI, USA). Library construction was performed with the NEXTFLEX Rapid DNA-Seq Kit, and sequencing was carried out on the MiSeq PE300 platform.

### 2.4. High-Throughput Sequencing Data Analysis

The quality of double-ended raw sequences was controlled using Fastp (0.19.6). Sequence splicing was performed with FLASH (v1.2.11), and noise reduction of optimized sequences was completed with the DADA2 plugin in Qiime2. Chloroplast and mitochondrial sequences were removed to minimize the number of retained sequences. Each sample demonstrated an average sequence coverage of 99.09% after splicing. Operational taxonomic units (OTUs) were analyzed for species taxonomy based on the SILVA 16S rRNA gene database (v138) using the Naive Bayes classifier in Qiime2. OTUs were plotted for Pan/Core analysis with the Vegan (v2.4.3) package in R (v3.3.1). Mothur (v1.30.2) was used to calculate the alpha diversity index, and between-group differences were tested with the Wilcoxon rank-sum test. The similarity of microbial community structures between samples was evaluated using PCoA based on the Bray–Curtis distance algorithm. Differences in microbial communities between sample groups were assessed with the PERMANOVA non-parametric test. LEfSe analysis (LDA > 4, *p* < 0.05) was applied to identify bacterial taxa with significant differences in abundance at levels ranging from phylum to genus. Microbial function was predicted with PICRUSt2 (v2.2.0). The results of intergroup diversity analyses and intergroup divergent species were corrected for multiple testing using the FDR method.

## 3. Results

### 3.1. Pan–Core Species Analysis

Pan refers to the total number of species across all samples (Figure 1A). Species in calves fed CON, OEO, and YCs gradually leveled off and plateaued, whereas the number of species in calves fed MIX continued to rise with increasing sample numbers. This indicates that additional species could be observed in calves fed MIX by increasing sample numbers. Core represents the number of species shared among all samples (Figure 1B). Core species gradually plateaued with increasing sample size across all treatments.

### 3.2. Venn Species Analysis

Calves fed CON demonstrated 680 OTUs and 78 endemic OTUs (Figure 2A, Venn species diagram). In comparison, OEO-fed calves exhibited 884 OTUs and 121 endemic OTUs, YC-fed calves demonstrated 888 OTUs and 161 endemic OTUs, and MIX-fed calves exhibited 2618 OTUs and 1730 endemic OTUs. The Average variation degree (AVD) represents bacterial community stability. There were no significant differences in rumen flora stability among the four treatments (Figure 2B).

### 3.3. Alpha Diversity Analysis

Calves fed MIX exhibited significantly greater (*p* < 0.05) Chao1 species richness (Figure 3A) compared to calves fed other treatments. However, the species coverage index (Figure 3B) and lineage diversity Pd index (Figure 3D) were similar (*p* > 0.05) among all treatments. The coverage index, which exceeded 0.99, indicates that current sequencing accurately represents bacterial groups. Calves fed MIX also exhibited significantly greater (*p* < 0.05) Shannon community diversity index values compared to calves fed other treatments (Figure 3C).

### 3.4. Beta Diversity Analysis

Calves fed MIX exhibited a higher degree of differentiation (*p* = 0.001 and R = 0.3981) in the microbial community compared with calves fed the other treatments (Figure 4). An overlap was observed among calves fed CON, OEO, and YCs, showing no significant separation.

### 3.5. Analysis of Species Composition

The top 10 species in relative abundance at the phylum level included *p_Firmicutes*, *p_Actinobacteriota*, *p_Bacteroidota*, *p_ Patescibacteria*, *p_Proteobacteria*, *p_Desulfobacterota*, *p_Cyanobacteria*, *p_unclassified_k_norank_d_Bacteria*, *p_Chloroflexi*, and *p_ Spirochaetota*. For calves fed MIX, *p_Actinobacteriota*, *p_Patescibacteria*, *p_Desulfobacterota*, *p_unclassified_k_norank_d_Bacteria*, and *p_Spirochaetota* were more abundant (*p* < 0.05) compared with calves fed the other treatments (Figure 5A; Table 2).

The top 10 species in relative abundance at the genus level were *g_norank_f_Eubacterium_coprostanoligenes_group*, *g_Olsenella*, *g_Lachnospiraceae_NK3A20_group*, *g_Erysipelotrichaceae_UCG-002*, *g_Acetitomaculum*, *g_Ruminococcus_gauvreauii_group*, *g_Bifidobacterium, g_norank_f_norank_o_Clostridia_UCG-014*, *g_Ruminococcus*, and *g_Eubacterium_nodatum_group*. Among these, *g_norank_f_Eubacterium_coprostanoligenes_group*, *g_Olsenella*, *g_Erysipelotrichaceae_UCG-002*, *g_Bifidobacterium* and *g_Ruminococcus* showed significant differences in relative abundance among the groups (*p* < 0.05) (Figure 5B; Table 3).

### 3.6. Species Difference Analysis

The LESfe screening method identified 26 biomarkers from the four treatments based on the LDA score at ≥4 (*p* < 0.05). Calves fed CON had 10 biomarkers, calves fed OEO had 5, calves fed YCs had 3, and calves fed MIX had 8 biomarkers. Biomarkers in calves fed CON, OEO, and YCs were primarily from *p_Firmicutes* and *p_Actinobacteriota*. In contrast, calves fed MIX had 8 biomarkers, mainly from *p_Patescibacteria*, *p_Bacteroidota*, and *p_Firmicutes* (Figure 6).

### 3.7. Species Functional Prediction Analysis

The functional ruminal microbial communities were predicted using PICRUSt2. Screening for differential functional pathways identified 19 pathways at KEEG Pathway level 2 (Figure 7). Calves fed CON exhibited the highest enrichment in infectious diseases (bacterial and parasitic), transcription, and the digestive system. Calves fed OEO showed greater activity in immune and nervous system functions. Calves fed YCs demonstrated increased activities in cellular community, prokaryote, and substructure functions. Calves fed MIX showed elevated activities in nucleotide metabolism, lipid metabolism, glycan biosynthesis and metabolism, amino acid metabolism, terpenoids and polyketides metabolism, drug resistance, antimicrobial processes, xenobiotic biodegradation and metabolism, antineoplastic processes, excretory systems, and exceptions to antineoplastic resistance (Figure 7; Table 4).

## 4. Discussion

### 4.1. Effect of OEO and YCs on Ruminal Microbial Diversity

Calves fed OEO and YCs showed negligible effects on species richness in the rumen when these products were fed separately. However, species richness increased when OEO and YCs were fed in combination. The OEO and YC combination not only increased species richness but also tended to enhance species diversity, as measured by Shannon’s index. When analyzing species composition, particular attention should be paid to the variation at the phylum level. In the MIX treatment, the abundance of Bacteroidota was significantly greater than in the other treatments. The abundance of Firmicutes in MIX was slightly higher compared to the other groups. Meanwhile, the abundance of Actinobacteria in MIX was the lowest among all treatments. The phylum of thick-walled bacillus (Firmicutes) mainly produces butyric acid, assisting intestinal mucosal repair and improving the intestinal barrier [23]. The phylum Bacteroidota, which mainly produces acetic acid and propionic acid, can inhibit cholesterol production and prevent metabolic diseases [24]. Firmicutes and Bacteroidetes are the most abundant phyla and core microorganisms in the digestive tract of ruminants, typically accounting for more than 90% of the abundance [25]. In this study, neither Firmicutes nor Bacteroidota reached the 90% level in CON, OEO, and YCs. In MIX, as Actinobacteria abundance declined, Bacteroidota abundance increased, and the abundance of both approached the 90% level. This result is critical for understanding the role of gastrointestinal homeostasis. In addition, PCoA identified a significant separation between the species composition of calves fed MIX and those fed the remaining treatments. This finding indicates a positive impact of MIX on the modulation of ruminal microbial composition. The OEO and YC combination enhanced ruminal microbial abundance and diversity compared to feeding OEO and YCs separately.

### 4.2. Ruminal Microbial Species Differences in Calves Fed OEO and YCs

Through Lefse analysis, ten differential biomarkers were identified for CON-fed calves, including *p_Actinobacteriota*, *c_Actinobacteria*, *o_Bifidobacteriales*, *f_Bifidobacteriaceae*, and *g_Bifidobacterium*, all of which belong to *p_Actinobacteriota*. *Actinobacteria* are filamentous Gram-positive bacteria that form branching structures at specific filament developmental stages. These bacteria can occur as spores or nutrients in various habitats, such as soil, aquatic environments, plant litter, compost, and food associated with plants, animals, and humans [26]. We identified *g_Bifidobacterium* as a marker of difference at the minimal taxonomic level. *Bifidobacterium bifidum* is a beneficial bacterium among actinomycetes [27]. It is an anaerobic Gram-positive bacillus that often bifurcates at the end, hence its name. *g_Bifidobacterium* is essential to the intestinal tract flora in humans and animals [28]. It is the second most abundant bacterium found in breastfed infants. Reduced numbers of *bifidobacteria* have been linked to various diseases, including obesity, diabetes, and allergies, at different life stages [29]. *Bifidobacteria* can alleviate digestive problems, improve glycemic control, lower lipid levels, boost immunity, exhibit antioxidant activity, help prevent eczema, and relieve stress and allergies [30,31,32]. Additionally, the remaining five species identified in CON-fed calves belong to *p_Firmicutes*, a large group of mostly Gram-positive bacteria with globular or rod-shaped thick cell walls. Many species in this phylum are beneficial, including core ruminant bacteria [33] such as *Lactobacillus, Lactobacillus brucei*, and *Ruminalococcus* [24].

Three differential biomarkers, including *g_Erysipelotrichaceae_UCG-009*, *g_norank_f_Eubacterium_coprostanoligenes_group*, and *g_Ruminococcus*, were identified at the minimal taxonomic level. *g_ Erysipelotrichaceae_UCG-009* is thought to influence host metabolism and inflammatory diseases [34]. *g_norank_f_Eubacterium_coprostanoligenes_groupcanproduce* produces butyric and propionic acid from fermentation metabolites [35], while *g_ Ruminococcus* primary produces acetic acid and formic acid. These acids are essential for gastrointestinal tract energy metabolism and help maintain intestinal barrier function by releasing polysaccharides [36].

Five differential biomarkers were identified in calves fed OEO, including *c_Coriobacteriia*, *o_Coriobacteriales*, *f_Atopobiaceae*, and *g_Olsenella*, all belonging to *p_Actinobacteriota*, while *g_ Coprococcus* belongs to *p_Firmicutes.* Increased *Olsenella* abundance promotes and elevates IL-10, an anti-inflammatory cytokine [37]. In contrast, *g_ Coprococcus*, a genus within the thick-walled bacterium phylum *Tricholobacteriaceae*, is an important intestinal organism [38]. Most *Tricholobacteriaceae* strains are isolated from feces and actively ferment carbohydrates to produce butyric acid [39]. *Coprococcus* bacteria are microbial biomarkers used to assess gastrointestinal tract health [40]. These bacteria may contribute to immune suppression and reduced allergic reactions [41].

Three biomarkers were identified in YC-fed calves, including *g_Roseburia* and *g_Catenisphaera* from *p_Firmicutes*, and *g_Nocardiopsis* from *p_Actinobacteriota. g_Roseburia* is associated with short-chain fatty acid production [42], while *g_Catenisphaera* may alleviate weaning stress in lambs [43]. *g_Nocardiopsishas* demonstrates strong bacteriostatic ability [44].

In MIX-fed calves, eight biomarkers were identified, including *f_Rikenellaceae* from *p_Bacteroidota*, *g_Rikenellaceae_RC9_gut_group*, and *g_Erysipelotrichaceae_UCG-002* from *p_Firmicutes*, as well as *c_Saccharimonadia* and *o_Patescibacteria* from *p_Patescibacteria*. *Saccharimonadales*, *f_Saccharimonadaceae*, and *g_Candidatus_Saccharimonas*, all members of *Rikenellaceae_RC9_gut_group*, were positively correlated with volatile fatty acid (VFA) production, regulating muscle fat deposition [45]. *g_Erysipelotrichaceae_UCG_002* is closely related to VFA synthesis [46], while *g_Candidatus_Saccharimonas* helps regulate intestinal homeostasis [47].

### 4.3. OEO and YC Impacts Ruminal Microbial Function

In this experiment, 18 pathways with significant differences were screened to predict microbial community function. Pathways significantly enriched in OEO included the immune system and nervous system. Previous research indicates that OEO promotes rumen epithelial development, enhancing nutrient digestion and utilization while improving immunity [48]. In human medicine, essential oils have shown efficacy against psychiatric disorders, including anxiolytic, antidepressant, sedative, and anticonvulsant effects [49]. These effects are attributed to active ingredients, such as alkenes, phenols, and alcohols, which influence the central nervous system of the organism [50].

Pathways significantly enriched in YCs included Cellular community—prokaryotes and Substance dependence. Yeast cells influence cellular processes, aligning with earlier findings [51]. However, the significant enrichment of Substance dependence remains unexplained. It is hypothesized that the observed enrichment may relate to disease development during that stage, influenced by the treatment drug.

There were numerous pathways significantly enriched in MIX that are central to this study. These included Nucleotide metabolism, Lipid metabolism, Glycan biosynthesis and metabolism, Metabolism of other amino acids, Metabolism of terpenoids and polyketides, Drug resistance: antimicrobial, Xenobiotics biodegradation and metabolism, Drug resistance: antineoplastic, and Excretory system. These pathways relate to metabolism, antimicrobial activity, and digestive systems. Previous studies demonstrated that OEO has such effects [52], while YCs promote metabolism, enhance immunity [53], and generate bioactive peptides with antimicrobial properties [54].

Compared with the addition of OEO or YCs alone, the effect of mixing the two substances on calf rumen microbiota was more pronounced. The species diversity of rumen microorganisms in calves increased significantly, while the stability of the flora remained unaffected. The abundance of Bacteroidota increased, and that of Actinobacteria decreased. The dominant species in the rumen shifted to Bacteroidota and Firmicutes, with the combined abundance of these two species exceeding 90% of the total microbial abundance. This shift positively influenced the stability of the rumen microbial environment in calves [55]. Furthermore, the combined addition of OEO and YCs had a more significant effect on rumen microorganisms than either substance alone. Analysis of microbial function revealed that the mixture enriched a greater number of functional pathways compared to OEO or YCs alone. This result indicates a higher presence of functional microorganisms in the group receiving the mixture. These microorganisms primarily supported amino acid metabolism, antimicrobial activity, and digestion, reflecting the unique biological properties of oregano essential oil and yeast culture [52,56]. The findings are also consistent with differential marker analysis, which identified more key biomarkers in the mixed-fed group. These biomarkers contributed to short-chain fatty acid production and gastrointestinal tract stability. This observation supports earlier findings that rumen microbes break down and utilize indigestible fibers to provide 70% of energy and 60–85% of amino acids to the host [57]. Amino acids, critical for healthy calf growth, are regulated by the body’s microbes and play an essential role in immunity and antioxidant capacity [58].

From the above results, we can conclude that feeding a combination of OEO and YCs to calves regulates rumen microorganisms more effectively than feeding either substance individually. The combined use of these substances produced no antagonistic effects and proved beneficial for microbial community regulation. Early ruminal development is critical for dairy and beef cattle production during the transition from liquid to dry feed [59]. Adequate nutritional supplementation, along with nutrient digestion, absorption, utilization, and efficiency, forms the basis for optimal calf growth, health, and oxidative status [57,58]. Rumen microorganisms play a crucial role in facilitating these processes [60,61,62].

## 5. Conclusions

Newborn Simmental crossbred male calves were fed OEO, YCs, and their combination as part of the calf starter diet. Rumen samples were collected on day 70, and the microbial community was analyzed through high-throughput sequencing to evaluate composition and changes. Feeding calves the OEO and YC combination resulted in greater richness, diversity, and species composition, along with differential biomarker effects. Microbial community functions were also enhanced. This combination proved more effective than feeding OEO or YCs alone.

## Figures and Tables

**Figure 1 animals-14-03710-f001:**
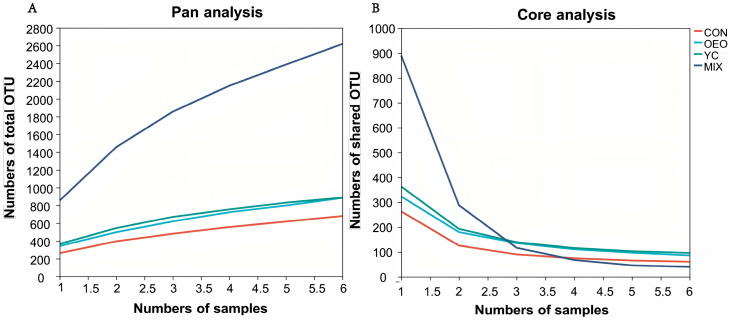
Pan–core species analysis. OTUs, or operational taxonomic units, cluster valid sequences obtained from sequencing, and cluster clean tags into OTUs at a default given similarity (default 97%). (**A**) Pan curves refer to pan OTUs and show how the number of all OTUs included in different samples varies as the number of samples increases. (**B**) Core curves refer to core OTUs and show how the number of shared OTUs present in different samples varies as the number of samples increases.

**Figure 2 animals-14-03710-f002:**
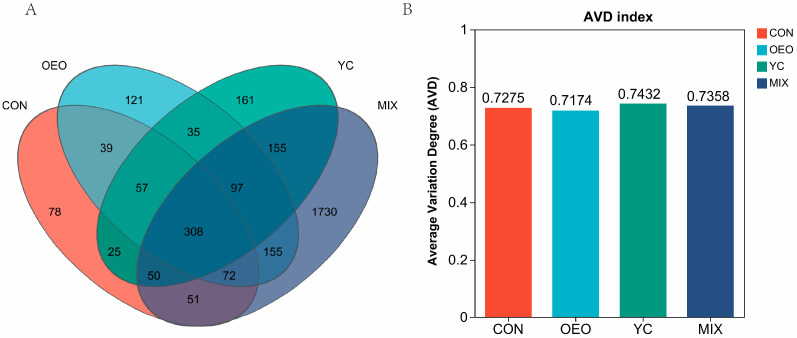
(**A**) Venn species diagram. (**B**) Average variability of bacterial communities. Different colors represent different subgroups. The same color blocks represent different subgroups, overlapping sections represent the number of species common to the other groups, non-overlapping sections represent the number of species common to the various groups, and non-overlapping sections represent the number of species common to the different groups.

**Figure 3 animals-14-03710-f003:**
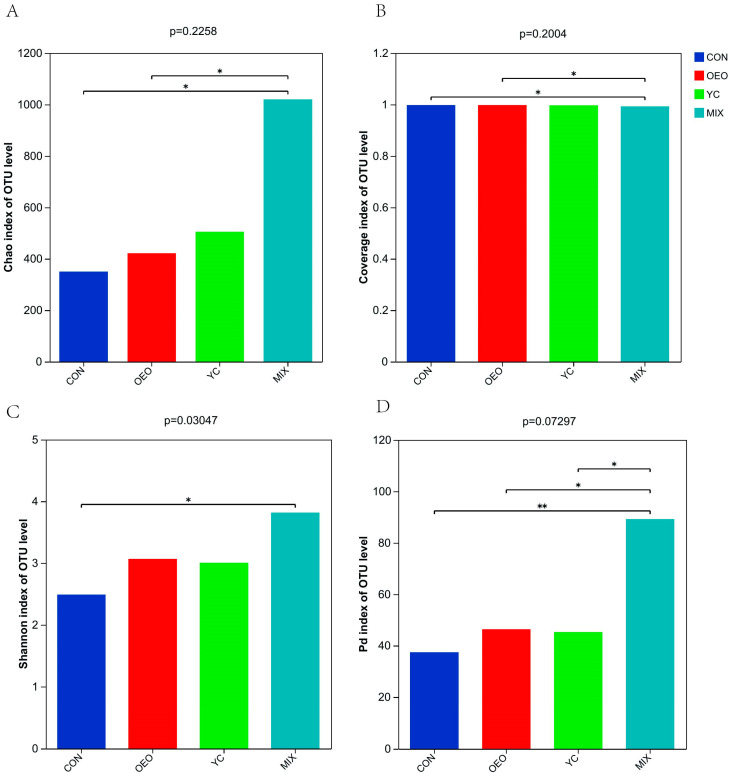
Species Alpha Diversity: (**A**) Chao1 index: representing species richness. (**B**) Coverage index: representing species coverage. (**C**) Shannon index: representing species diversity. (**D**) Species lineage diversity Pd index: representing species lineage diversity. In the figure, * indicates a significant difference, ** indicates a highly significant difference.

**Figure 4 animals-14-03710-f004:**
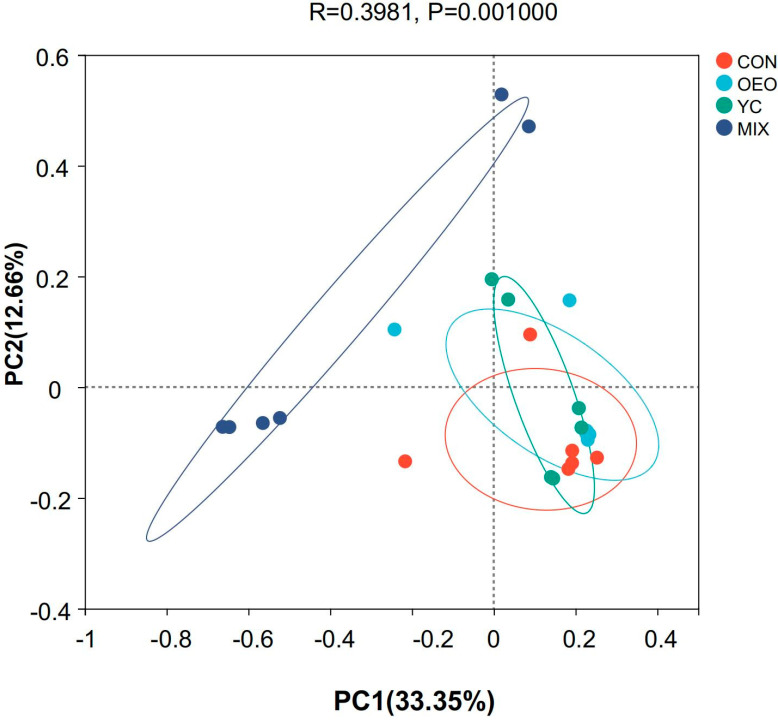
Species Beta Diversity: PCoA analysis and Principal coordinates were analyzed, with distances between colored circles representing similarities or differences in community composition between groups.

**Figure 5 animals-14-03710-f005:**
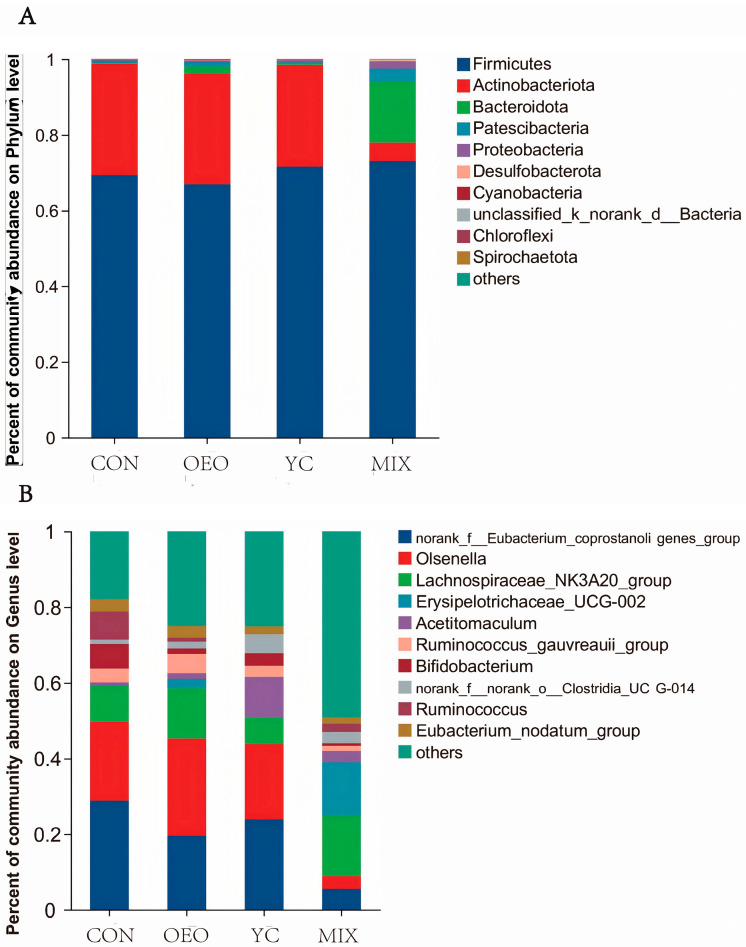
Species composition at different levels (Top 10). The vertical coordinate is the proportion of species abundance in that sample, with different colored bars representing different species and the length of the bar representing the size of that species: (**A**) phylum-level species; (**B**) genus-level species.

**Figure 6 animals-14-03710-f006:**
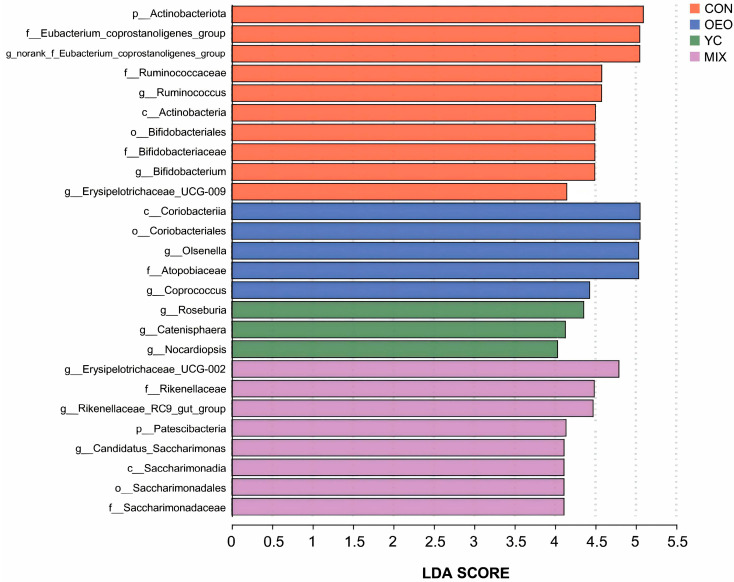
Analysis of species differences. The horizontal coordinate is the LDA value, and the vertical coordinate is the species name, with larger LDA scores representing a greater influence of species abundance on the differential effect.

**Figure 7 animals-14-03710-f007:**
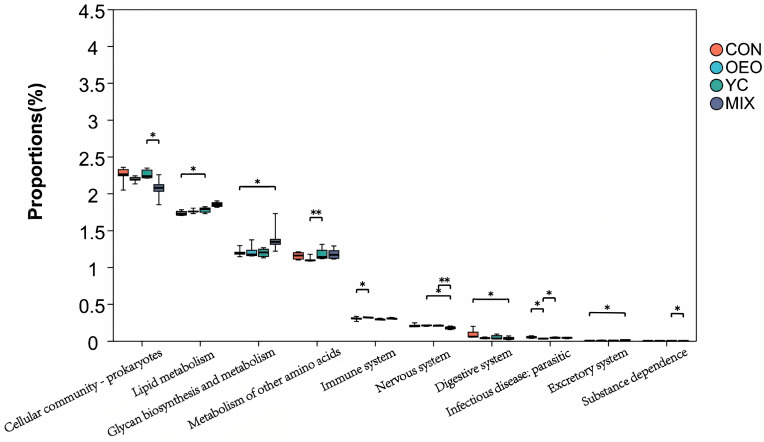
Species function prediction analysis. * means the difference is significant (*p* < 0.05); ** means the difference is highly significant (*p* < 0.01).

**Table 1 animals-14-03710-t001:** Calf starter ingredient and nutrient composition (dry matter basis, %).

Ingredient Composition	Content	Nutrient Level	Content
Corn	40.54	DM	87.95
Soybean meal	32.00	CP	22.17
Wheat bran	5.80	EE	3.79
Cottonseed meal	5.30	Ash	5.91
Puffed soybeans	5.00	ADF	6.18
Whey powder	4.00	NDF	12.23
Molasses	4.00	Ca	0.91
CaCO_3_	1.60	P	0.59
Soybean oil	0.80		
NaCl	0.60		
CaHPO_4_	0.10		
MgO	0.10		
Selenium yeast	0.02		
Premix	0.14		
Total	100.00		

DM: Dry matter; CP: Crude protein; EE: Ether extract; Ash: Crude ash; ADF: Acid detergent fiber; NDF: Neutral detergent fiber; Ca: Calcium; P: Phosphorus.

**Table 2 animals-14-03710-t002:** Species composition of rumen microorganisms at Phylum levels in calves (%).

Species at the Phylum Level	Group				*p*-Value
	CON	OEO	YC	MIX	
p__Firmicutes	69.36 ± 17.90	66.92 ± 9.38	71.6 ± 16.32	73.08 ± 18.18	0.7932
p__Actinobacteriota	29.48 ± 18.40a	29.32 ± 12.69a	26.86 ± 16.42a	4.852 ± 5.36b	0.0127
p__Bacteroidota	0.1877 ± 0.25	1.855 ± 3.98	0.4534 ± 0.266	16.1 ± 21.71	0.0916
p__Patescibacteria	0.4485 ± 0.55bc	1.057 ± 0.63b	0.2978 ± 0.22bc	3.452 ± 2.36a	0.0024
p__Proteobacteria	0.3545 ± 0.15	0.424 ± 0.40	0.5298 ± 0.32	1.952 ± 3.30	0.7181
p__Desulfobacterota	0.03956 ± 0.04b	0.1272 ± 0.18ab	0.08768 ± 0.06b	0.2967 ± 0.11a	0.0182
p__Cyanobacteria	0.08928 ± 0.07	0.1604 ± 0.29	0.1219 ± 0.09	0.02513 ± 0.02	0.0673
p__unclassified_k__norank_d__Bacteria	0.03208 ± 0.03ab	0.008554 ± 0.01b	0.03154 ± 0.04ab	0.124 ± 0.10a	0.0362
p__Chloroflexi	0.00695 ± 0.01	0.0695 ± 0.04	0.006415 ± 0.01	0.01497 ± 0.02	0.0933
p__Spirochaetota	0.003208 ± 0.00b	0.02727 ± 0.06b	0.002673 ± 0.01b	0.05774 ± 0.04a	0.0393

Note: CON refers to control group; OEO refers to oregano essential-oil-treated group; YC refers to yeast-treated group; MIX refers to oregano essential oil and yeast combination addition group. a, b, c Means within the same row with unlike superscripts differ, *p* < 0.05.

**Table 3 animals-14-03710-t003:** Species composition of rumen microorganisms at Genus levels in calves (%).

Species at the Genus Level	Group				*p*-Value
	CON	OEO	YC	MIX	
*g__norank_f__Eubacterium_coprostanoligenes_group*	28.83 ± 16.12a	19.55 ± 7.12a	23.91 ± 8.51a	5.547 ± 4.27b	0.009
*g__Olsenella*	20.95 ± 13.25a	25.68 ± 13.00a	19.88 ± 17.11a	3.341 ± 4.64b	0.021
*g__Lachnospiraceae_NK3A20_group*	9.468 ± 10.81	13.35 ± 5.13	7.013 ± 5.18	16.02 ± 6.57	0.087
*g__Erysipelotrichaceae_UCG-002*	0.1171 ± 0.12ab	2.396 ± 5.43ab	0.0005346 ± 0.00c	14.05 ± 21.88a	0.011
*g__Acetitomaculum*	0.7725 ± 0.71	1.533 ± 1.50	10.76 ± 14.71	3.011 ± 3.04	0.209
*g__Ruminococcus_gauvreauii_group*	3.582 ± 3.85	5.13 ± 3.33	2.921 ± 1.41	1.38 ± 1.12	0.249
*g__Bifidobacterium*	6.532 ± 5.36a	1.45 ± 2.06ab	3.344 ± 1.50ab	0.6175 ± 0.97b	0.007
*g__norank_f__norank_o__Clostridia_UCG-014*	1.134 ± 1.74	1.759 ± 0.67	4.979 ± 5.33	3.02 ± 2.13	0.209
*g__Ruminococcus*	7.417 ± 18.01a	1.053 ± 1.94ab	0.0139 ± 0.01b	2.247 ± 2.08ab	0.002
*g__Eubacterium_nodatum_group*	3.298 ± 2.70	3.073 ± 1.14	2.044 ± 2.01	1.601 ± 1.95	0.523

Note: CON refers to control group; OEO refers to oregano essential-oil-treated group; YC refers to yeast-treated group; MIX refers to oregano essential oil and yeast combination addition group. a, b, c Means within the same row with unlike superscripts differ, *p* < 0.05.

**Table 4 animals-14-03710-t004:** Differential microbial function prediction pathways (%).

Microbial Function	Group				*p*-Value
	CON	OEO	YC	MIX	
Translation	3.865 ± 0.10a	3.80 ± 0.03ab	3.708 ± 0.05b	3.782 ± 0.08ab	0.018
Nucleotide metabolism	2.769 ± 0.07b	2.78 ± 0.04b	2.735 ± 0.03b	2.819 ± 0.04a	0.033
Cellular community—prokaryotes	2.25 ± 0.11ab	2.19 ± 0.04ab	2.263 ± 0.06a	2.065 ± 0.13b	0.019
Lipid metabolism	1.733 ± 0.03b	1.76 ± 0.02ab	1.776 ± 0.04ab	1.847 ± 0.04a	0.002
Glycan biosynthesis and metabolism	1.198 ± 0.05ab	1.209 ± 0.09ab	1.194 ± 0.06b	1.384 ± 0.18a	0.024
Metabolism of other amino acids	1.152 ± 0.05ab	1.103 ± 0.04b	1.178 ± 0.08ab	1.179 ± 0.07a	0.029
Metabolism of terpenoids and polyketides	0.954 ± 0.03ab	0.9749 ± 0.01ab	0.9398 ± 0.04b	0.9981 ± 0.03a	0.040
Drug resistance: antimicrobial	0.7952 ± 0.02b	0.7861 ± 0.05ab	0.8238 ± 0.06ab	0.9074 ± 0.04a	0.009
Infectious disease: bacterial	0.7248 ± 0.05a	0.7143 ± 0.03ab	0.6688 ± 0.02b	0.7236 ± 0.03ab	0.044
Xenobiotics biodegradation and metabolism	0.6429 ± 0.08ab	0.6314 ± 0.05b	0.6496 ± 0.01ab	0.7472 ± 0.11a	0.032
Immune system	0.3 ± 0.02ab	0.3156 ± 0.00a	0.2895 ± 0.01b	0.3012 ± 0.01ab	0.007
Drug resistance: antineoplastic	0.2113 ± 0.01ab	0.2168 ± 0.01ab	0.2194 ± 0.00ab	0.2493 ± 0.02a	0.005
Nervous system	0.2054 ± 0.02a	0.2065 ± 0.01a	0.2056 ± 0.01a	0.1772 ± 0.02b	0.009
Transcription	0.1933 ± 0.01a	0.1891 ± 0.00ab	0.1837 ± 0.00ab	0.1806 ± 0.01b	0.001
Digestive system	0.0916 ± 0.06a	0.0392 ± 0.01ab	0.0494 ± 0.03ab	0.0374 ± 0.02b	0.049
Infectious disease: parasitic	0.0493 ± 0.01a	0.0316 ± 0.00b	0.0433 ± 0.01ab	0.0406 ± 0.01ab	0.011
Excretory system	0.0019 ± 0.00ab	0.0036 ± 0.00ab	0.0042 ± 0.00a	0.009 ± 0.01b	0.015
Substance dependence	0.0004 ± 0.00b	0.0004 ± 0.00b	0.0011 ± 0.00a	0.0004 ± 0.00b	0.040

Note: CON refers to control group; OEO refers to oregano essential-oil-treated group; YC refers to yeast-treated group; MIX refers to oregano essential oil and yeast combination addition group. a, b Means within the same row with unlike superscripts differ, *p* < 0.05.

## Data Availability

Please contact the corresponding author for data access.

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
