# Peer review of "Effects of Oregano Essential Oil and/or Yeast Cultures on the Rumen Microbiota of Crossbred Simmental Calves"

_animals, 2024, doi:10.3390/ani14243710_

Round 1
Reviewer 1 Report (Previous Reviewer 2)
Comments and Suggestions for Authors
The manuscript by Liu et al. investigates the effects of supplementing calves with oregano essential oil, yeast culture, and their combination. The study finds that mixed supplementation significantly enhances rumen microbial richness, species diversity, and functional metabolic pathways, outperforming the individual supplements.
In my previous review of this paper, I noted that it "holds potential interest because the formal analysis appears well-conducted, despite some limitations, and the results are noteworthy." However, I also expressed concerns that the paper's strengths were undermined by poor writing, inadequate data presentation, and insufficient discussion. At that time, I recommended that the authors "consider hiring a professional for a thorough rewrite," which they did with the addition of David P. Casper’s contribution. His input has substantially improved the manuscript, but I still have a few minor recommendations:
Line 23: Change "might be more promoting of" to "might better promote."
Lines 111-112: Replace "feed conversions" and "environmentally friendly" with "feed conversion" and "environmental sustainability."
Line 116: Change "microflora" to "microbiota."
Line 234: Change "high" to "higher."
Lines 388-390: Please rewrite this sentence for better clarity.
Author Response
I sincerely thank you for your valuable comments and suggestions. Your professional insights are extremely important to me, and I have carefully read and thoughtfully considered all the feedback.
I have carefully considered each suggestion and made corresponding modifications and improvements in the article.
Reviewer 2 Report (New Reviewer)
Comments and Suggestions for Authors
Why six animals per treatment was chosen? The sample size justification is missing, which weakens the validity of results, particularly for a microbial study. Most importantly, in the experimental treatments, authors did not use antibiotic as experimental treatment and whole introduction based on antibiotics replacement with additives. Therefore, it is advised that authors must use antibiotic in treatment to check the effect of oregano essential oil (OEO) and yeast culture (YC) compare to antibiotics. In this case, control is missing and manuscript is rejected. Authors are advised to change the problem statement and rewrite the whole manuscript in context of oregano essential oil (OEO) and yeast culture (YC) as feed additive rather as alternative of antibiotics.
Simple summary is too long, authors are advised to reduce it into 2-3 lines. Authors should not express previous studies results rather to focus on their research.
Abstract
Line 25-28: unnecessary details. Please remove it. Oregano essential oil (OEO) is a natural plant extract possessing antibacterial, antiviral, antifungal, and antioxidant characteristics. In contrast, yeast culture (YC) is produced via anaerobic fermentation on a specific medium that when fed to livestock improves nutrient digestibility while promoting gastrointestinal health via microbial colonization and competitive exclusion.
Line 45-46: only provide results in this section ‘These path ways focus on the metabolism of various substances, antimicrobial, and digestive systems’
Introduction
I have found flaws in flow and clarity. The introduction lacks logical flow, with some sections jumping between topics without clear transitions. Background information on antibiotic use and alternative additives is extensive. From the introduction section, It is unclear why oregano essential oil (OEO) and yeast culture (YC) were selected over other feed additives.
Most importantly, in the experimental treatments, authors did not use antibiotic as experimental treatment and whole introduction is based on alternative antibiotics. Therefore, it is advised that authors must use antibiotic in treatment to check the effect of oregano essential oil (OEO) and yeast culture (YC) compare to antibiotics. In this case, control is missing and manuscript is rejected.
Authors are advised to change the problem statement and rewrite the whole manuscript in context of oregano essential oil (OEO) and yeast culture (YC) as feed additive rather as alternative of antibiotics.
Materials and Methods
Why six animals per treatment was chosen? The sample size justification is missing, which weakens the validity of results, particularly for a microbial study. Additionally, the randomization process is not clearly described.
Discussion
The discussion is primarily descriptive and lacks insight into the mechanisms behind the observed microbial changes. The discussion lacks information on the implications of changes in specific microbial populations. For example, what does an increase in Rikenellaceae mean for rumen health? Addressing this would provide more depth.
Comments on the Quality of English Language
Need to improve
Author Response
Thank you for taking your valuable time to review our article and for recognizing our work. We have rewritten and rechecked the article based on the suggestions you made. Through your review report, we also found the shortcomings of our work, which also provided us with better ideas and methods for our future research work. Thank you again for your work. We have tried our best to reply to your questions, so we hope you can review them.
Simple summary is too long, authors are advised to reduce it into 2-3 lines. Authors should not express previous studies results rather to focus on their research.
Response : we removed some unnecessary paragraphs. The introduction of previous studies in it is also our own work. I think this part of the introduction makes sense for the subsequent work in this article.
Abstract
Line 25-28: unnecessary details. Please remove it. Oregano essential oil (OEO) is a natural plant extract possessing antibacterial, antiviral, antifungal, and antioxidant characteristics. In contrast, yeast culture (YC) is produced via anaerobic fermentation on a specific medium that when fed to livestock improves nutrient digestibility while promoting gastrointestinal health via microbial colonization and competitive exclusion.
Line 45-46: only provide results in this section ‘These path ways focus on the metabolism of various substances, antimicrobial, and digestive systems’
Response : We deleted the description of L25-28. Also the description of L45-46 appears only in the abstract because the results of the focus are summarized in this section, which is described in detail in the discussion section.
Introduction
I have found flaws in flow and clarity. The introduction lacks logical flow, with some sections jumping between topics without clear transitions. Background information on antibiotic use and alternative additives is extensive. From the introduction section, It is unclear why oregano essential oil (OEO) and yeast culture (YC) were selected over other feed additives.
Most importantly, in the experimental treatments, authors did not use antibiotic as experimental treatment and whole introduction is based on alternative antibiotics. Therefore, it is advised that authors must use antibiotic in treatment to check the effect of oregano essential oil (OEO) and yeast culture (YC) compare to antibiotics. In this case, control is missing and manuscript is rejected.
Authors are advised to change the problem statement and rewrite the whole manuscript in context of oregano essential oil (OEO) and yeast culture (YC) as feed additive rather as alternative of antibiotics.
Response : Based on your comments, we have rewritten the introductory section of the text.
Materials and Methods
Why six animals per treatment was chosen? The sample size justification is missing, which weakens the validity of results, particularly for a microbial study. Additionally, the randomization process is not clearly described.
Response : The number of head of cattle was restricted to the farm where the trial was conducted. The sample size for microbial sequencing was at least 3. Compared to this 6 is a reasonable sample size. We also preformed the analysis by first analyzing the reasonableness of the sequencing results and the depth of sequencing. In the Figure 1 section of the text, the results show that continuing to increase the sample size does not significantly increase the number of species obtained from sequencing. The sequencing results from this experiment can be analyzed. It was only after obtaining this result that we proceeded with the subsequent work.
Discussion
The discussion is primarily descriptive and lacks insight into the mechanisms behind the observed microbial changes. The discussion lacks information on the implications of changes in specific microbial populations. For example, what does an increase in Rikenellaceae mean for rumen health? Addressing this would provide more depth.
Response : Based on your comments, we've added a bit of new information to the discussion section.
Reviewer 3 Report (New Reviewer)
Comments and Suggestions for Authors
Effects of oregano essential oil and/or yeast cultures on the rumen microbiota of crossbred Simmental calves
The manuscript was based on an interesting study assessing the effect that supplementing oregano essential oil and (OEO)/or yeast culture (YC) has on the rumen microbiome of pre-weaned calves. The authors have done a good job in better understanding how feeding OEO and YC combination improved the modulation of the rumen microbial community in pre-weaned calves compared with calves fed without or with OEO and YC fed separately. However, I have few comment where I feel the manuscript could be improved, especially in the results and discussion sections.
In the research design, why did the authors decide to use a model based on beef calves nursed by their dams instead of a dairy calve model?
Methods: no need for subheading 2.3
Lines 158-162: Confusing sentence, rewrite for clarity.
Results: Some of the results sections are confusing and not clearly presented. The analysis for the differences in abundance from phylum to genus level, was called in the results section "species differences analysis". The differences observed at different taxonomical classifications were called biomarkers. Some of the identified biomarkers belong to the same bacteria across the different taxonomy classification, for example: g_Bifidobacterium, f_Bifidobacteriaceae, o_Bifidobacteriales, c_Actinobacteria. Additionally, the subheading species functional prediction analysis is no clearly presented. See further comments in attached document.
Discussion. Some sections of the discussion needs further documentation or citations to enrich this. For example, the part that discussed the effects of the feeding additives on the ruminal microbial diversity. In this subheading of the discussion the authors did not cite previous findings from their authorship mentioned in the introduction. See comments in attached document
The conclusions are supported by the results

Author Response
Thank you for reviewing our article and pointing out the deficiencies in our work, which is crucial for us to improve the quality of our article and conduct future research. We have responded to and revised the questions you raised, and the changes and additions are marked in red in the text in the hope that they will be recognised by you, and thank you once again for your work for us.
Methods: In the research design, why did the authors decide to use a model based on beef calves nursed by their dams instead of a dairy calve model?
Response:In China, the early weaning technique for beef calves is being demonstrated and promoted with the aim of enhancing the reproductive rate of beef cows. Against such an beef production backdrop, in the collaborating cattle farms, all the calves are housed in individual pens, which can fully meet the requirements of the experiment. Moreover, compared with dairy calves, there have been fewer experiments on novel additives such as essential oils, yeast culture and probiotics conducted on beef calves. Consequently, we have chosen to carry out this research on beef calves.
Methods: no need for subheading 2.3
Lines 158-162: Confusing sentence, rewrite for clarity.
Response:Subheading 2.3 has been deleted from the article
Results: Some of the results sections are confusing and not clearly presented. The analysis for the differences in abundance from phylum to genus level, was called in the results section "species differences analysis". The differences observed at different taxonomical classifications were called biomarkers. Some of the identified biomarkers belong to the same bacteria across the different taxonomy classification, for example: g_Bifidobacterium, f_Bifidobacteriaceae, o_Bifidobacteriales, c_Actinobacteria. Additionally, the subheading species functional prediction analysis is no clearly presented. See further comments in attached document.
Response:The results section has been modified.
Discussion. Some sections of the discussion needs further documentation or citations to enrich this. For example, the part that discussed the effects of the feeding additives on the ruminal microbial diversity. In this subheading of the discussion the authors did not cite previous findings from their authorship mentioned in the introduction. See comments in attached document
Response:We have carefully revised the discussion section.
Reviewer 4 Report (New Reviewer)
Comments and Suggestions for Authors
Effects of oregano essential oil and/or yeast cultures on the rumen microbiota of crossbred Simmental calves by Ting Liu
The paper aimed to evaluate the influence of the supplementation of OEO and YC alone or in combination on the ruminal microbiota development. The authors hypothesized that combining OEO and yeast would modulate the rumen microbiota for promoting gastrointestinal homeostasis and function.
The paper is very well written; the authors performed the trial using adequate methodologies and the results are very well depicted. Also the conclusions are consistent with the results.
Anyway few corrections and/or additions need, as follows:
line 66: please provide the meaning of GDP
line 90: please move (OEO) to line 87 close to oregano essential oil
line 130: please add OEO close the amount administered to this group, as you did for YC group
line 140; what is the meaning of @ ??
line 141: please write d 15 – 28 because treatment was the same (2x/d for 2 hours)
line 409: after [56] please add the sentence: “as well as the oxidative status (Musco et al., 2020)”
References
please add the following:
Musco, N.; Tudisco, R..; Grossi, M.; Mastellone, V.; Morittu, V.M.; Pero, M.E.; Wanapat, M.; Trinchese, G.; Cavaliere, G.; Mollica, M.P.; et al. Effect of a high forage: Concentrate ratio on milk yield, blood parameters and oxidative status in lactating cows. Ani. Prod. Sci. 2020, 60, 1531–1538.
Author Response
I sincerely thank you for your valuable comments and suggestions on my paper. Your professional insights are extremely important to me, and I have carefully read and thoughtfully considered all the feedback. I have given careful consideration to each suggestion and made corresponding modifications and improvements in the article. Here is my detailed response to your valuable feedback:
line 66: Gross Domestic Product(GDP),In China, the value of all final goods and services produced in the economy of a country or region in a quarter or a year.
line 90: Modifications have been made.
line 130: Changes have been made as requested.
line 140: The @ character has been removed from this line.
line 141: This has been amended to read d 15 – 28.
line 409: Modifications have been made.
References added.
Thank you to the reviewer for recommending relevant literature, which has provided valuable references and insights for the study. I have appropriately cited these references in the article and updated the reference section accordingly. This not only strengthens the academic background of my paper but also enhances the depth and breadth of the research.
Reviewer 5 Report (New Reviewer)
Comments and Suggestions for Authors
You need to correct all the manuscript on spaces between words and number of citation. Or final point and first word of the next phrase. There is a lot of missing spaces.
Ttitle of the manuscript do not represent the work. The work focus on the immune system, evaluating the microbiotic ruminal population, however, this not seems to be the focus of the work. The authors evaluated the OEO administration with YC on rumen microbiotic, however do not evaluated the animals development as effect of the rumen development and also did not evaluated the rumen wall development. Which is interligated with the rumen colonization and function.
Simple summary: It should have the description of the oregano' component that enhance animals health, beside the description of essential oil. Explaining how this compound could replace the antibiotics. Like the first line of the abstract.
L27: I think that medium is not the correct term to be used in here
L30-31: Please, include the body weight of the animals and the average age. Newborn by itself do not categorize the animals.
L32-34: Need more information about how the treatments were offered/administrated to the animals, the caracteristicas of the feed used.
L50: Reorder the keywords to alphabetical order
L65: GDP, what that means?
Introduction: 3 points: first, I suggest you to include some data on cases that proves the bacterial resistence due to antibiotic use. Second, you could explain in a more depth way the essencial oil action on animals health. The activation of immune system, how it works, how this increase fight the diseases. Third, what is the actual hyphothesis of administrating the oregano essential oil or yeast to the animals? Only promote the increase in immune system without the antibiotic use? Or could be a second point? Once yeast could also improve rumen microbiotic developtment?
L124: Newborn, could we understand as the were born in the day that started the experiment? Also, the animals received the diet with or without additives as soon as the were born? How? In trough?
L140: ad libitum must be in italic
Item 2.2 is a little bit confusing
Title of tables do not have final point
Tables: You need to standardize the amount of numbers used in p-value and other values
Results: Is a little bit to big and confusing
Discussion: Seems like a continuation of results. There is a great amount of literature citation to support the results, however, do not discuss the results. The part more interesting as discussion, is the last topic, mostly in the L399 to L409.
L416-417: This is the aim in the conclusion? Why?
Author Response
Thank you very much for taking the time to review our article. We apologize for our carelessness in writing and we have revised the text for grammatical errors. Based on your comments, we have revised the problems in the text, and the revisions are marked in red in the text. We hope you can review our revisions and thank you again for your time.
Simple summary:Corrections have been made as requested.
L27:Changes have been made in the appropriate places.
L30-31:We have noted in the text (first day of life,birth weight ≥ 35 kg).
L32-34:We have added more detailed information, including in 2.1 Test animals and 2.2 Feeding procedure.
L50:Keywords have been arranged as required.
L65:Gross Domestic Product(GDP),In China, the value of all final goods and services produced in the economy of a country or region in a quarter or a year.
Introduction:We have changed the text accordingly.
L124:We have revised calf birth and weight and related feeding programs in the article.
L140:ad libitum has been changed to italics.
Item 2.2 has been modified in the text.
The final point of the table title has been removed.
The p-value in the tables have been standardized as required.
Results:We have reworked it in the text.
Discussion:We finalized the discussion section as requested.
L416-417:We've reworked this unclear description already.
Round 2
Reviewer 2 Report (New Reviewer)
Comments and Suggestions for Authors
Authors did not follow instruction for example
Simple summary is too long, authors are advised to reduce it into 2-3 lines. Authors should not express previous studies results rather to focus on their research.
Simple summary is still too long
Abstract
I mentioned that ‘only provide results in this section’ authors still have unnecessary information in this section
Introduction
Please follow previous comments
Most importantly, in the experimental treatments, authors did not use antibiotic as experimental treatment and whole introduction is based on alternative antibiotics. Therefore, it is advised that authors must use antibiotic in treatment to check the effect of oregano essential oil (OEO) and yeast culture (YC) compare to antibiotics. In this case, control is missing and manuscript is rejected.
Authors are advised to change the problem statement and rewrite the whole manuscript in context of oregano essential oil (OEO) and yeast culture (YC) as feed additive rather as alternative of antibiotics.
Build hypothesis, objective and clearly state problem
English language is too poor to understand which disturb reading of the manuscript. Please improve language before sending to reviewer
Comments on the Quality of English Language
Too poor. improve it before sending to reviewers please
Author Response
Reviewer 2
1.Comments and Suggestions:Authors did not follow instruction for example
Simple summary is too long, authors are advised to reduce it into 2-3 lines. Authors should not express previous studies results rather to focus on their research.
Simple summary is still too long
Response : I would like to express my sincere gratitude for your valuable comments and suggestions. I have carefully revised the Simple Summary in accordance with your requirements. The new Simple Summary is as follows:
Feeding calves a mixture of oregano essential oil and yeast cultures led to increased rumen microbial richness and diversity, along with regulated relative abundances of particular species. Moreover, pathways associated with metabolism and antimicrobials were enriched. The research indicates that this mixed additive outperforms separate feeding of oregano essential oil and yeast culture in modulating the rumen microbial community of calves.
- Comments and Suggestions:
Abstract
I mentioned that ‘only provide results in this section’ authors still have unnecessary information in this section
Response : Based on the four essential elements of writing: purpose, method, result, and conclusion, as well as your valuable suggestions, I have revised the abstract section again and removed the information unrelated to the results.
The new abstract is as follows:
This study hypothesized that combining oregano essential oil (OEO) and yeast cultures (YC) would modulate rumen microbiota to promote gastrointestinal homeostasis and function. Twenty-four newborn, healthy, disease-free, crossbred Simmental male calves (birth weight ≥ 35 kg) were assigned to one of four treatments based on birth data. Treatments were: 1) Control (CON), calves fed calf starter without additives; 2) OEO, calves fed calf starter containing 60 mg/kg body weight (BW) of OEO per day; 3) YC, calves fed calf starter containing 45 mg/kg BW of YC per day; and 4) MIX, calves fed calf starter with OEO (60 mg/kg, BW) and YC (45 mg/kg, BW) combination. The experimental period lasted 70 days. Rumen fluid was collected on the final day, and 16S rRNA sequencing was performed to assess alterations in rumen microbiota. Calves fed MIX exhibited significantly greater microbial richness, species diversity, and lineage diversity (p < 0.05) compared with calves in the other groups. MIX-fed calves also showed changes (p < 0.05) in the relative abundance of certain rumen species, identified as biomarker through LEfSe analysis (LDA > 4, p < 0.05). These biomarkers included f_Rikenellaceae, g_Rikenellaceae_RC9_gut_group, g_Erysipelotrichaceae_UCG-002, c_Saccharimonadia, o_Saccharimonadales, f_Saccharimonadaceae, and g_Candidatus_Saccharimonas. Pathways enriched (p < 0.05) in Mix-fed calves involved nucleotide metabolism, lipid metabolism, glycan biosynthesis and metabolism, amino acid metabolism, terpenoids and polyketides metabolism, antimicrobial drug resistance, xenobiotic biodegradation and metabolism, antineoplastic drug resistance and excretory system pathways. In conclusion, this study demonstrates that the OEO and YC combination enhances rumen microbial community modulation in calves more effectively than OEO or YC fed individually or with the control diet.
- Comments and Suggestions:
Introduction
Please follow previous comments
Most importantly, in the experimental treatments, authors did not use antibiotic as experimental treatment and whole introduction is based on alternative antibiotics. Therefore, it is advised that authors must use antibiotic in treatment to check the effect of oregano essential oil (OEO) and yeast culture (YC) compare to antibiotics. In this case, control is missing and manuscript is rejected.
Authors are advised to change the problem statement and rewrite the whole manuscript in context of oregano essential oil (OEO) and yeast culture (YC) as feed additive rather as alternative of antibiotics.
Build hypothesis, objective and clearly state problem
Response : Your suggestions are excellent and I fully agree. Therefore, I have rewritten the entire introduction. The new introduction is as follows:
Newborn calves play a crucial role in livestock farming due to their early contribution to rumen microbiota establishment, which affects feed digestion, energy conversion, growth, and later production performance and health. In light of the global antibiotic ban, the development of safe feed additives to enhance rumen microbiota composition has become critical. Studies show that plant essential oils, yeast cultures (YC), antimicrobial peptides, Lactobacillus, and organic acids can serve as alternatives to antibiotics, directly benefiting calf health. Oregano essential oil (OEO), a natural plant extract, is widely used as a feed additive for its antibacterial, antiviral, antifungal, and antioxidant properties. Previous studies by our team demonstrated that OEO enhances calf growth and immune function and modulates rumen microbiota to strengthen immunity. Research also indicates that OEO increases Lactobacillus abundance in piglets, reduces Enterobacteriaceae, and enriches rumen cocci, bifidobacteria, and enterococci in sheep. It also boosts metabolites such as indole-3-acetic acid and indole aldehyde, improving growth and intestinal barrier function. Although OEO shows promise in promoting livestock growth and refining rumen microbiota composition, most studies focus on its standalone use. Limited research addresses potential synergistic or antagonistic effects when OEO is combined with other feed additives. Thus, investigating combinations that synergistically enhance rumen microbiota in calves together with OEO remains essential.
YC are produced by anaerobic fermentation and are subsequently dried on specific carriers under tightly controlled production conditions. These include yeast cells, yeast metabolites, and components of the culture medium. YC are widely used in animal husbandry. Studies indicate that supplementing YC can improve cattle growth performance, enhance rumen development, and increase the abundance of fiber-degrading bacteria, lactic acid-utilizing bacteria, and carbohydrate-degrading bacteria. This leads to improved rumen function and a better feed conversion rate. Adding active yeast can accelerate the maturation of the rumen microbiota in lambs and stabilize the rumen environment. In a recent study, supplementation with YC increased the abundance of non-fiber-degrading bacteria in Tibetan sheep while reducing pathogenic bacteria in the rumen. This suggests that yeast not only supports rumen microbiota stability but also strengthens immunity. In a previous study, we tested a combination of OEO and sodium butyrate but found no significant effects. Given that YC can influence the rumen microbiota of ruminants, we chose to combine yeast with OEO. Currently, literature on the combined use of natural feed additives in ruminants is scarce, but limited studies suggest that specific combinations can improve ruminant performance. This study aims to evaluate the effects of OEO and YC, used either alone or in combination, on the development of the rumen microbiota. The hypothesis is that combining OEO and YC will regulate the rumen microbiota and promote gastrointestinal balance and function.
- Comments and Suggestions:
English language is too poor to understand which disturb reading of the manuscript. Please improve language before sending to reviewer
Comments on the Quality of English Language
Too poor. improve it before sending to reviewers please
Response : Thank you for your suggestions. English writing can indeed be a matter of different opinions. Although I have already invited Professor Dave Casper from North Carolina A&T State University to revise the language of the whole paper, it still failed to meet your language requirements. Therefore, in this round, I sent the paper to company for language modification and obtained an editorial certificate. Please check the attachment. I hope it can meet your requirements. Thank you again for your suggestions.

Reviewer 5 Report (New Reviewer)
Comments and Suggestions for Authors
Ok.
Author Response
Thank you for your suggestions. English writing can indeed be a matter of different opinions. Although I have already invited Professor Dave Casper from North Carolina A&T State University to revise the language of the whole paper, it still failed to meet your language requirements. Therefore, in this round, I sent the paper to company for language modification and obtained an editorial certificate. Please check the attachment. I hope it can meet your requirements. Thank you again for your suggestions.

Round 3
Reviewer 2 Report (New Reviewer)
Comments and Suggestions for Authors
Thanks for revision. Manuscript is improved greatly
This manuscript is a resubmission of an earlier submission. The following is a list of the peer review reports and author responses from that submission.
Round 1
Reviewer 1 Report
Comments and Suggestions for Authors
Animals-3173129-peer-review-v1 [Review Comments]
Line 25: Delete "newborn" to avoid repetition.
Line 23: Replace "and the mixture of the two" with "and their combination" or "and their mixture."
Lines 29–31: Delete this sentence as it repeats information from Lines 28–29.
Line 29: Specify the amount of each additive combined for the mixed group: is it half (30 mg OEO & 2.5 g YCG) or the exact amount (60 mg OEO & 5 g YCG) used for individual treatment?
Line 96: The author mentions "A large number of studies....." but cites only one reference (No. 18). Please add more relevant references.
Line 114: Replace "butyric acid" with "yeast culture" since the study focuses on yeast culture, not butyric acid.
Line 119: Specify the amount of each additive combined for the mixed group used, as previously stated in Line 29.
Lines 144–146: The statement should read: "DNA was extracted according to the product instructions using the E.Z.N.A.® Soil DNA Kit (Omega B144 io-tek, Norcross, GA, USA), and tested for …. … … …"
Lines 166–170: Recast this sentence in the past tense. For example: "Fastp (0.19.6) was used for quality control of double-ended raw sequencing data..."
Line 201: Venn diagram should be “Fig. (2)”, not “Fig. 1B”.
Line 210-211: Change Fig. 2A to Fig. 3A, Fig. 2D to Fig. 3D, and Fig. 2E to Fig. 3C.
Line 215: Insert the reference “Fig. 3C”.
Lines 297–298: This appears to be an incomplete statement. Revise to: "The effect on microbial diversity was not stated."
Line 306–307: Move this sub-heading (4.2) to the next line.
Line 315: Italicize the scientific name (Bifidobacterium bifidum) and all other scientific names throughout the manuscript.
Author Response
Thank you for reviewing our thesis, for your efforts to express our gratitude, our research and the article has many shortcomings, your opinion is an important factor in our learning and growth, we have been based on your comments on the paper of the problems in the revision, such as there are still errors and shortcomings, but also please can point out that, once again, thank you for your work.
Line 25: Delete "newborn" to avoid repetition.
Reply:Modifications have been made.
Line 23: Replace "and the mixture of the two" with "and their combination" or "and their mixture."
Reply:Modifications have been made.
Lines 29–31: Delete this sentence as it repeats information from Lines 28–29.
Reply:Modifications have been made.
Line 29: Specify the amount of each additive combined for the mixed group: is it half (30 mg OEO & 2.5 g YCG) or the exact amount (60 mg OEO & 5 g YCG) used for individual treatment?
Reply:Modifications have been made.
Line 96: The author mentions "A large number of studies....." but cites only one reference (No. 18). Please add more relevant references.
Reply:Thank you for your guidance, this conclusion was reached through our previous research article, which was misrepresented in translation and has been corrected.
Line 114: Replace "butyric acid" with "yeast culture" since the study focuses on yeast culture, not butyric acid.
Reply:Modifications have been made.
Line 119: Specify the amount of each additive combined for the mixed group used, as previously stated in Line 29.
Reply:Modifications have been made.
Lines 144–146: The statement should read: "DNA was extracted according to the product instructions using the E.Z.N.A.® Soil DNA Kit (Omega B144 io-tek, Norcross, GA, USA), and tested for …. … … …"
Reply:Modifications have been made.
Lines 166–170: Recast this sentence in the past tense. For example: "Fastp (0.19.6) was used for quality control of double-ended raw sequencing data..."
Reply:Modifications have been made.
Line 201: Venn diagram should be “Fig. (2)”, not “Fig. 1B”.
Reply:Modifications have been made.
Line 210-211: Change Fig. 2A to Fig. 3A, Fig. 2D to Fig. 3D, and Fig. 2E to Fig. 3C.
Reply:Modifications have been made.
Line 215: Insert the reference “Fig. 3C”.
Reply:Modifications have been made.
Lines 297–298: This appears to be an incomplete statement. Revise to: "The effect on microbial diversity was not stated."
Reply:Modifications have been made.
Line 306–307: Move this sub-heading (4.2) to the next line.
Reply:Modifications have been made.
Line 315: Italicize the scientific name (Bifidobacterium bifidum) and all other scientific names throughout the manuscript.
Reply:Modifications have been made.
Reviewer 2 Report
Comments and Suggestions for Authors
The manuscript by Liu et al. investigates the effects of supplementing calves with oregano essential oil, yeast culture, and their combination. The study finds that the mixed supplementation significantly enhances rumen microbial richness, species diversity, and functional metabolic pathways, outperforming the effects of the individual supplements.
This study holds potential interest because the formal analysis appears well-conducted, even though with some limitations, and the results are noteworthy. However, these strengths are undermined by poor writing, inadequate data presentation, and insufficient discussion.
The primary limitation in the study's design mirrors the concern raised by reviewer Woo-Hyun Kim regarding your team's previously published paper in Animals (https://doi.org/10.3390/ani14060820): “The individual calf has unique microbiota at birth and develops them over time, influenced by factors such as the source of colostrum and the cow’s condition. To better elucidate the effect of OEO and changes in microbiota due to OEO supplementation, the authors should have analyzed the community before feeding them as well.” Although you indicated that this would be addressed in future studies, you have again only analyzed the microbiota after the trial’s conclusion. This limitation is even more critical in the present study. Additionally, you have not specified whether the calves were fed colostrum, and it remains unclear what they were fed, as described in lines 129 and 130.
In addition to this design limitation, the writing quality is extremely poor, with an alarming number of typographical errors, repeated sentences or words, and overall confusing phrasing. A few examples include:
Line 19: You failed to mention that yeast culture was added to the MIX group.
Lines 19-31: This sentence repeats the preceding sentences.
Line 93: “Microcosmic” should be “microscopic.”
Lines 114-115: The yeast culture group is incorrectly named the “butyric acid group.”
Line 162: “Fluorometer (Promega)” is repeated.
Line 201: It’s Figure 2, not “1B.”
Lines 210-216: “Figures 2A, 2D, 2E” should be referenced as Figures 3A, 3D, and 3E, respectively.
Line 236: “Top five 10” should be “top 10.”
Lines 277-278: The phrase “in the CON group” is duplicated.
Line 345: “[335]” should be “[35].”
Lines 377-379: The paragraph is confusing and includes the repeated phrase “metabolism, metabolism of other amino acids, metabolism of other amino acids, and metabolism of other amino acids. amino acids.”
Beyond the issues with English and typographical errors, I have some specific concerns:
Simple Summary: The summary only states the aim of the study, omitting “pertinent results, conclusions from the study, and how they will be valuable to society,” as required by the journal’s guidelines.
Abstract: In addition to the issues already mentioned, the abstract lacks an introductory sentence discussing oregano essential oil and yeast culture supplementation in calves, which should precede the study’s aim.
Materials and Methods: Please clarify whether the yeast culture supplementation is 5 g per day or 5 g per hour, as there is a contradiction between the abstract and line 118 (“5g/h”). Additionally, specify what the calves were fed in lines 129 and 130. Table 1 is not referenced in the text and lacks explanations of the units or the meanings of abbreviations used.
Results: The statement in line 194, “the MIX group had the highest number of shared species,” is accurate only for the analysis of the first three samples according to Figure 1B. The figures are all very small and of poor quality, making analysis difficult. Define “OTUs” before using the term in line 196. Add “Fig. 3C” at the end of line 216. Section 3.5 is very confusing; although you analyze relative abundance at the phylum and genus levels, you incorrectly refer to “species.” Just to refer to some of the many places you have to change, the first column of Table 2 should be labeled “Phylum name,” and Table 3 should be labeled “Genus name.” The statement in lines 234-235, “species composition of each group of microorganisms was similar at the phylum and genus levels,” is incorrect based on your results, which show statistically significant differences between groups. The first column of Table 4 should be “Microbial function.”
Discussion: In addition to poor English and typographical errors, the discussion is overly focused on human medicine, with inadequate analysis of the significance of your results for calves.
Conclusion: As with the rest of the paper, the conclusion should be rewritten to be clearer, with shorter sentences and improved English. Replace “microflora” with “microbiota,” and revise the phrases “a more microbial community” and “Enrichment was also richer” for clarity.
Comments on the Quality of English LanguageThe English quality is very poor, with basic and confusing sentences throughout. Professional editing is necessary to improve clarity and readability.
Author Response
Thank you very much that you can review our thesis, there are many shortcomings in our work, your suggestions are very good to help us, we have revised the thesis according to your suggestions, if there are still shortcomings and mistakes, please point them out again, thank you again for your help to us.
The manuscript by Liu et al. investigates the effects of supplementing calves with oregano essential oil, yeast culture, and their combination. The study finds that the mixed supplementation significantly enhances rumen microbial richness, species diversity, and functional metabolic pathways, outperforming the effects of the individual supplements.
This study holds potential interest because the formal analysis appears well-conducted, even though with some limitations, and the results are noteworthy. However, these strengths are undermined by poor writing, inadequate data presentation, and insufficient discussion.
The primary limitation in the study's design mirrors the concern raised by reviewer Woo-Hyun Kim regarding your team's previously published paper in Animals (https://doi.org/10.3390/ani14060820): “The individual calf has unique microbiota at birth and develops them over time, influenced by factors such as the source of colostrum and the cow’s condition. To better elucidate the effect of OEO and changes in microbiota due to OEO supplementation, the authors should have analyzed the community before feeding them as well.” Although you indicated that this would be addressed in future studies, you have again only analyzed the microbiota after the trial’s conclusion. This limitation is even more critical in the present study. Additionally, you have not specified whether the calves were fed colostrum, and it remains unclear what they were fed, as described in lines 129 and 130.
In addition to this design limitation, the writing quality is extremely poor, with an alarming number of typographical errors, repeated sentences or words, and overall confusing phrasing. A few examples include:
Reply:We did consider the issue of collecting rumen fluid from newborn calves for microbial sequencing in subsequent experiments. Still, after collecting relevant information, the volume of the actual stomach is the largest in newborn calves, accounting for about 70% of the total. In contrast, the rumen, reticulum, and valvular stomachs are very small, accounting for only 30% of the total. Their functions are imperfect, so the rumen does not function as the main digestive and absorptive organ in newborn calves, and it gradually functions when the calf is more than one month old. The rumen gradually functions when calves are one month old or older. The rumen fluid cannot be collected at birth, and it is more challenging to collect it at one month of age. It must be collected several times in many days, significantly influencing calves' health condition. The farm's production process does not allow our experimenters to operate this way. We have not yet found a reasonable way to collect some of the earlier rumen fluid to compare with the results at the end of the trial without harming the calves, and we would be grateful for any advice you can give us.
Line 19: You failed to mention that yeast culture was added to the MIX group.
Reply:Changes have been made to the article mix group additions.
Lines 19-31: This sentence repeats the preceding sentences.
Reply: The error here has been removed.
Line 93: “Microcosmic” should be “microscopic.”
Reply:Modifications have been made.
Lines 114-115: The yeast culture group is incorrectly named the “butyric acid group.”
Reply:Modifications have been made.
Line 162: “Fluorometer (Promega)” is repeated.
Reply: Deletions have been made.
Line 201: It’s Figure 2, not “1B.”
Reply:Modifications have been made.
Lines 210-216: “Figures 2A, 2D, 2E” should be referenced as Figures 3A, 3D, and 3E, respectively.
Reply:Modifications have been made.
Line 236: “Top five 10” should be “top 10.”
Reply:Modifications have been made.
Lines 277-278: The phrase “in the CON group” is duplicated.
Reply:Modifications have been made.
Line 345: “[335]” should be “[35].”
Reply:Modifications have been made.
Lines 377-379: The paragraph is confusing and includes the repeated phrase “metabolism, metabolism of other amino acids, metabolism of other amino acids, and metabolism of other amino acids. amino acids.”
Reply:Modifications have been made.
Beyond the issues with English and typographical errors, I have some specific concerns:
Simple Summary: The summary only states the aim of the study, omitting “pertinent results, conclusions from the study, and how they will be valuable to society,” as required by the journal’s guidelines.
Reply: The brief summary section has been modified to add results and conclusions as well as roles.
Abstract: In addition to the issues already mentioned, the abstract lacks an introductory sentence discussing oregano essential oil and yeast culture supplementation in calves, which should precede the study’s aim.
Reply: We added a brief description of the two additives in the abstract section, before the purpose of the study.
Materials and Methods: Please clarify whether the yeast culture supplementation is 5 g per day or 5 g per hour, as there is a contradiction between the abstract and line 118 (“5g/h”). Additionally, specify what the calves were fed in lines 129 and 130. Table 1 is not referenced in the text and lacks explanations of the units or the meanings of abbreviations used.
Reply:We have modified the representation of additive quantities by citing Table 1 and describing the abbreviations therein.
Results: The statement in line 194, “the MIX group had the highest number of shared species,” is accurate only for the analysis of the first three samples according to Figure 1B. The figures are all very small and of poor quality, making analysis difficult. Define “OTUs” before using the term in line 196. Add “Fig. 3C” at the end of line 216. Section 3.5 is very confusing; although you analyze relative abundance at the phylum and genus levels, you incorrectly refer to “species.” Just to refer to some of the many places you have to change, the first column of Table 2 should be labeled “Phylum name,” and Table 3 should be labeled “Genus name.” The statement in lines 234-235, “species composition of each group of microorganisms was similar at the phylum and genus levels,” is incorrect based on your results, which show statistically significant differences between groups. The first column of Table 4 should be “Microbial function.”
Reply: Based on your comments, we have revised and corrected the presentation of the table header, the similarity of microorganisms at the phylum level, and the figure notes.
Discussion: In addition to poor English and typographical errors, the discussion is overly focused on human medicine, with inadequate analysis of the significance of your results for calves.
Reply:We have corrected the article for English language errors and added to the overall discussion of the results.
Conclusion: As with the rest of the paper, the conclusion should be rewritten to be clearer, with shorter sentences and improved English. Replace “microflora” with “microbiota,” and revise the phrases “a more microbial community” and “Enrichment was also richer” for clarity.
Reply:We provide a new description of the concluding section.
Thank you very much that you can review our thesis, there are many shortcomings in our work, your suggestions are very good to help us, we have revised the thesis according to your suggestions, if there are still shortcomings and mistakes, please point them out again, thank you again for your help to us.
The manuscript by Liu et al. investigates the effects of supplementing calves with oregano essential oil, yeast culture, and their combination. The study finds that the mixed supplementation significantly enhances rumen microbial richness, species diversity, and functional metabolic pathways, outperforming the effects of the individual supplements.
This study holds potential interest because the formal analysis appears well-conducted, even though with some limitations, and the results are noteworthy. However, these strengths are undermined by poor writing, inadequate data presentation, and insufficient discussion.
The primary limitation in the study's design mirrors the concern raised by reviewer Woo-Hyun Kim regarding your team's previously published paper in Animals (https://doi.org/10.3390/ani14060820): “The individual calf has unique microbiota at birth and develops them over time, influenced by factors such as the source of colostrum and the cow’s condition. To better elucidate the effect of OEO and changes in microbiota due to OEO supplementation, the authors should have analyzed the community before feeding them as well.” Although you indicated that this would be addressed in future studies, you have again only analyzed the microbiota after the trial’s conclusion. This limitation is even more critical in the present study. Additionally, you have not specified whether the calves were fed colostrum, and it remains unclear what they were fed, as described in lines 129 and 130.
In addition to this design limitation, the writing quality is extremely poor, with an alarming number of typographical errors, repeated sentences or words, and overall confusing phrasing. A few examples include:
Reply:We did consider the issue of collecting rumen fluid from newborn calves for microbial sequencing in subsequent experiments. Still, after collecting relevant information, the volume of the actual stomach is the largest in newborn calves, accounting for about 70% of the total. In contrast, the rumen, reticulum, and valvular stomachs are very small, accounting for only 30% of the total. Their functions are imperfect, so the rumen does not function as the main digestive and absorptive organ in newborn calves, and it gradually functions when the calf is more than one month old. The rumen gradually functions when calves are one month old or older. The rumen fluid cannot be collected at birth, and it is more challenging to collect it at one month of age. It must be collected several times in many days, significantly influencing calves' health condition. The farm's production process does not allow our experimenters to operate this way. We have not yet found a reasonable way to collect some of the earlier rumen fluid to compare with the results at the end of the trial without harming the calves, and we would be grateful for any advice you can give us.
Line 19: You failed to mention that yeast culture was added to the MIX group.
Reply:Changes have been made to the article mix group additions.
Lines 19-31: This sentence repeats the preceding sentences.
Reply: The error here has been removed.
Line 93: “Microcosmic” should be “microscopic.”
Reply:Modifications have been made.
Lines 114-115: The yeast culture group is incorrectly named the “butyric acid group.”
Reply:Modifications have been made.
Line 162: “Fluorometer (Promega)” is repeated.
Reply: Deletions have been made.
Line 201: It’s Figure 2, not “1B.”
Reply:Modifications have been made.
Lines 210-216: “Figures 2A, 2D, 2E” should be referenced as Figures 3A, 3D, and 3E, respectively.
Reply:Modifications have been made.
Line 236: “Top five 10” should be “top 10.”
Reply:Modifications have been made.
Lines 277-278: The phrase “in the CON group” is duplicated.
Reply:Modifications have been made.
Line 345: “[335]” should be “[35].”
Reply:Modifications have been made.
Lines 377-379: The paragraph is confusing and includes the repeated phrase “metabolism, metabolism of other amino acids, metabolism of other amino acids, and metabolism of other amino acids. amino acids.”
Reply:Modifications have been made.
Beyond the issues with English and typographical errors, I have some specific concerns:
Simple Summary: The summary only states the aim of the study, omitting “pertinent results, conclusions from the study, and how they will be valuable to society,” as required by the journal’s guidelines.
Reply: The brief summary section has been modified to add results and conclusions as well as roles.
Abstract: In addition to the issues already mentioned, the abstract lacks an introductory sentence discussing oregano essential oil and yeast culture supplementation in calves, which should precede the study’s aim.
Reply: We added a brief description of the two additives in the abstract section, before the purpose of the study.
Materials and Methods: Please clarify whether the yeast culture supplementation is 5 g per day or 5 g per hour, as there is a contradiction between the abstract and line 118 (“5g/h”). Additionally, specify what the calves were fed in lines 129 and 130. Table 1 is not referenced in the text and lacks explanations of the units or the meanings of abbreviations used.
Reply:We have modified the representation of additive quantities by citing Table 1 and describing the abbreviations therein.
Results: The statement in line 194, “the MIX group had the highest number of shared species,” is accurate only for the analysis of the first three samples according to Figure 1B. The figures are all very small and of poor quality, making analysis difficult. Define “OTUs” before using the term in line 196. Add “Fig. 3C” at the end of line 216. Section 3.5 is very confusing; although you analyze relative abundance at the phylum and genus levels, you incorrectly refer to “species.” Just to refer to some of the many places you have to change, the first column of Table 2 should be labeled “Phylum name,” and Table 3 should be labeled “Genus name.” The statement in lines 234-235, “species composition of each group of microorganisms was similar at the phylum and genus levels,” is incorrect based on your results, which show statistically significant differences between groups. The first column of Table 4 should be “Microbial function.”
Reply: Based on your comments, we have revised and corrected the presentation of the table header, the similarity of microorganisms at the phylum level, and the figure notes.
Discussion: In addition to poor English and typographical errors, the discussion is overly focused on human medicine, with inadequate analysis of the significance of your results for calves.
Reply:We have corrected the article for English language errors and added to the overall discussion of the results.
Conclusion: As with the rest of the paper, the conclusion should be rewritten to be clearer, with shorter sentences and improved English. Replace “microflora” with “microbiota,” and revise the phrases “a more microbial community” and “Enrichment was also richer” for clarity.
Reply:We provide a new description of the concluding section.
Reviewer 3 Report
Comments and Suggestions for Authors
Cannot review paper until properly prepared for manuscript with better English, grammar, and properly describing methods.
Comments on the Quality of English LanguagePoor English and grammar.
Author Response
Reply: Thank you for taking time out of your busy schedule to review our paper, because our native language is not English, and we are careless in writing, which has caused trouble to your work, we have revised the grammatical errors in the article, and there may still be mistakes, please can point out that we will certainly be the first time to make corrections, and once again, thank you for your work!
Round 2
Reviewer 1 Report
Comments and Suggestions for Authors
Dear Authors,
I have reviewed the edited manuscript and found that you have addressed the issues raised.
Thank you for the contribution to scientific knowledge.
Author Response
Dear reviewer
Greetings! Thank you very much for reviewing our thesis, your work is an affirmation that helps me to keep working hard. We sincerely hope that you are doing well and in good health!
Sincerely
Reviewer 2 Report
Comments and Suggestions for Authors
I appreciate that the authors have addressed the typographical errors I previously highlighted and have rewritten some sentences. However, the paper still requires substantial revisions and corrections. My initial comments were intended just to illustrate that the manuscript was not ready for publication and needs a comprehensive revision and rewrite.
I also acknowledge the authors' explanation of the limitations related to the unique microbiota of calves. However, this explanation should also be incorporated into the discussion section of the paper.
In addition to the writing issues, as mentioned in my previous review, the poor quality of several figures—specifically Figures 1, 5, 6, and 7—renders them unreadable. The image quality must be significantly improved before the paper can be considered for publication.
I will now point out additional writing issues to further illustrate again the need for a thorough rewrite. These examples are not exhaustive, as reviews are not meant to correct poor English and typographical errors in full.
Simple Summary
Line 17: “to reduce the use of antibiotics” instead of “as alternatives to antibiotics”
Abstract:
Line 34: remove “crucial”
Line 36: remove a dot before “(MIX)”
Line 37: Change to “A 70-day formal trial was conducted, and on the last day, rumen fluid was collected…”
Improve the writing of point “(2)” and remove italic from the words “three phylum level species, namely”
The point (3) must be rewritten as it is very confusing.
The abstract also lacks a portion of discussion and conclusion. The results should be summarized for those portions to be added.
Main text:
Line 56: not all antibiotics are “secondary metabolites of microorganisms”.
Paragraph of lines 70-72 confusing.
Line 85: remove “of the effect”
Line 88: don’t start the sentence only by “In short, it is”
Line 96: should be “so we chose another substance to combine with oregano essential oil for the combination study”
Improve the writing of lines 99-100
Change “many studies at home” of line 101
Line 104: change “lacta-tion-increasing” to “lactation-increasing”
Write in smaller and clearer sentences, improving the sentences from lines 103-109; 115-124; 149-164
Don’t start the sentence of line 182 by “analyse” and also change “analyse” in line 184.
Remove “use” in line 186.
Change the sentence of lines 198-199 to “The core is the number of species shared by all samples as can be seen in Figure 1B. The number of shared species…”
Line 219: change to “As shown in Figure 3.,”
Lines 233-234: change to “The results showed a partial overlap between CON, OEO, and YCG, with no significant separation observed.”
Change “wholly” to “significantly” in line: 235
Improve the clarity of point 3.7
Line 303. Change the words “with the help”
Line 304: should be changed to “…had a negligible effect on species richness in the rumen when given separately.”
Line 307: “Using” instead of "According to”
The sentence of the lines 307-309 as it is don’t make sense and all that paragraph is very repetitive.
Change “According to” in line 319
Improve the writing of sentences from lines 385-391.
Line 401: change to “… did not produce antagonistic effects and were beneficial…”
In summary, the manuscript requires substantial improvements in both writing and content. I strongly recommend that the authors consider hiring a professional to perform a thorough rewrite. Once the necessary revisions have been made, including the improvement of the quality of the figures, the authors may then resubmit the paper to this or another journal for reconsideration.
Comments on the Quality of English LanguageThe English report has already been included in the review above.
Author Response
Dear reviewer
Greetings! We have looked for professionals to rework our paper according to your comments. We have also revised the figures in the charts and graphs. Your review of our paper is very detailed and serious, we sincerely thank you for your efforts, we have already solved the problems you raised in the article, if we are still insufficient, please point out, once again, I wish you good luck in your work and good health!
Sincerely
Reviewer 3 Report
Comments and Suggestions for Authors
The authors did well at improving the English language issues that I had with the manuscript. The manuscript in its current state is much improved and I commend the authors for making the edits. After reanalyzing the paper I have no other comments to the authors.
Author Response
Dear reviewer
Thank you for your work, it is our guidepost to improve the quality of the paper. We wish you all the best in your work and good health!
Sincerely